# Association of Vitamin D Knowledge, Behavior and Attitude with BMI Status among Arab Adults

**DOI:** 10.3390/ijerph191711107

**Published:** 2022-09-05

**Authors:** Nasser M. Al-Daghri, Hanan Alfawaz, Nasiruddin Khan, Yousef Al-Saleh, Naji J. Aljohani, Dara Aldisi, Ghadah Alkhaldi, Amani M. Alqarni, Hadeel O. Almasoudi, Lina A. Alshehri, Rinad M. Alanzi, Malak N. K. Khattak, Mohamed A. Elsaid, Majed S. Alokail

**Affiliations:** 1Chair for Biomarkers of Chronic Diseases, Biochemistry Department, King Saud University, Riyadh 11451, Saudi Arabia; 2Department of Food Science & Nutrition, College of Food & Agriculture Science, King Saud University, Riyadh 11495, Saudi Arabia; 3Department of Food Science and Human Nutrition, College of Applied and Health Sciences, A’Sharqiyah University, Ibra 400, Oman; 4College of Medicine, King Saud bin Abdulaziz University for Health Sciences, Riyadh 22490, Saudi Arabia; 5King Abdullah International Medical Research Center, Riyadh 11481, Saudi Arabia; 6Department of Medicine, King Abdulaziz Medical City, Ministry of National Guard Health Affairs, Riyadh 11461, Saudi Arabia; 7Obesity Endocrine and Metabolism Center, King Fahad Medical City, Riyadh 11525, Saudi Arabia; 8Department of Community Health Sciences, College of Applied Medical Sciences, King Saud University, Riyadh 11451, Saudi Arabia; 9Protein Research Chair, Biochemistry Department, King Saud University, Riyadh 11451, Saudi Arabia

**Keywords:** vitamin D, BMI, knowledge, behavior, attitude, sun exposure, Saudi Arabia

## Abstract

This study aims to investigate the association of vitamin D (VD) knowledge, behavior, and attitude with BMI status among Saudi adults. This cross-sectional online survey included a total of 774 participants (M/F: 239/535). Knowledge about the overall sources of VD was highest in OB participants in correctly identifying sunlight (95.1%; *p* < 0.001) while significantly more OW participants answered food (83.1%; *p* = 0.04) and fortified food (66.5%; *p* = 0.02). However, 18.9% of OB participants also wrongly identified air as a VD source and this was significantly higher than in other groups (*p* = 0.03). OW participants were 50% less likely to identify salmon and fish oil (odds ratio, OR 0.5 (95% Confidence interval, CI 0.4–0.7); *p* < 0.01) and 40% more likely to identify chicken (OR 1.4 (1.0–1.9); *p* < 0.05) as dietary sources of VD than controls. On the other hand, OB participants were almost three times more likely to know that sunlight exposure is the main source of VD than controls (OR 2.65 (1.2–6.0); *p* < 0.05). In conclusion, while VD knowledge overall was apparently high in Saudi adults regardless of BMI status, the quality of knowledge among OB and OW individuals appear inconsistent, particularly in terms of identifying the right VD sources. Public health awareness campaigns should include the correction of VD misconceptions so that high-risk populations are able to make well-informed decisions in achieving optimal VD levels.

## 1. Introduction

The increasing prevalence of vitamin D (VD) deficiency, defined as circulating 25 hydroxyvitamin (OH) D < 50 nmol/L or <20 ng/mL, is now considered a major public health issue at a global level [1]. This is especially alarming in low- to middle-income countries where VD deficiency in adults ranges from 51–99% [2]. In the Middle East, particularly in Saudi Arabia where there is no shortage of sunlight, the 10-year (2008–2017) overall prevalence of VD deficiency for all age groups was 73.2% [3]. While a majority of VD is obtained through exposure to ultraviolet (UVB) sunlight, several dietary sources (e.g., salmon, egg yolks, mushrooms, and fortified dairy products) also contain VD [4]. What makes VD deficiency of substantial clinical concern has everything to do with its function in human metabolism. For many decades it is known that vitamin D (VD) has a pivotal role in the regulation of calcium and bone homeostasis [5]. In the last couple of years, however, accumulating literature consistently suggest that VD deficiency is significantly associated with a roster of major chronic diseases such as atherosclerosis, hypertension, diabetes, autoimmune diseases, cancer, cognition, and dental disorders [1,2,6,7,8,9,10,11].

In Saudi Arabia, several local studies have demonstrated the inverse relationship of VD with major cardiometabolic parameters [12,13,14]. Furthermore, emerging local evidence in Saudi Arabia also implicates VD deficiency with biological premature aging [15], domains of intelligence [16], muscle mass [17], and severe acute respiratory syndrome coronavirus 2 (SARS-CoV-2) infection [18,19]. Several factors lead to low VD deficiency in the Saudi population. The cultural practice of using abaya (traditional clothing that covers the whole body) among females limits the availability of exposed skin and hence hinders direct sunlight exposure. Moreover, people prefer indoor activities rather than outdoor physical activities due to extreme hot weather conditions [20,21]. A recent local study demonstrated a high level of awareness among students about the risk of unprotected sun exposure (such as sunburn, aging, and skin cancer), an opinion of using sunscreen as harmful, and wearing protective clothing as a protective means for direct sun exposure [22]. Aljefree and colleagues demonstrated low levels of knowledge among the Saudi adult population about VD along with low consumption of VD and calcium supplementation, as a major reason leading to the higher prevalence of VD deficiency among coronary heart disease patients [23]. Another study among Saudi adults revealed a scarcity of knowledge in relation to dietary sources of VD, with some participants showing negative attitudes toward sun exposure [24]. The lack of knowledge on VD coupled with lifestyle factors intrinsic to the culture predisposes several populations such as pregnant women and the elderly to adverse outcomes secondary to VD deficiency [25].

Given the available local literature provided, knowledge of one obvious risk factor for VD deficiency remains largely under investigation in the general Saudi community and that is obesity. Given that one out of every four Saudi adults is obese [26], it is interesting to investigate whether the community’s level of knowledge and perceptions of vitamin D is associated with their BMI status. The present study, therefore, aims to examine the relationship between the above-mentioned variables and BMI among the adult Saudi population.

## 2. Materials and Methods

### 2.1. The Study Design and Participants

The present cross-sectional study was designed to investigate the knowledge, behavior, and attitude of Saudi adult males and females (aged 18 years and above) with regard to VD and its association with BMI. The survey was conducted from 20 October to 8 December 2021. A questionnaire was designed and cascaded on social media apps such as WhatsApp groups and Twitter handles. One response per ID was allowed to ensure no duplication of data. The sample size was determined using the estimated population size (20,000) adults, accepted margin of error = 3.45%, level of confidence = 95%, and the response distribution rate = 50%, the actual sample size = 774 was obtained using Raisoft^®^ sample size calculator. For the purpose of this study, only Saudi adults versed in digital platforms such as WhatsApp and Twitter were invited to participate. Non-Saudis and those below 18 were excluded from the study. Ethical approval was obtained from the Ethics Committee of the College of Science Ref No: KSU-HE-21-379, King Saud University, Riyadh, Saudi Arabia. A consent form was included as the first part of the questionnaire and participants were required to agree before they can proceed to the next page.

### 2.2. Questionnaire, Data Collection, and Measurements

A pilot study (N = 100 participants) was performed to ascertain the test-retest reliability and validity of the questionnaire, guided by previous works on questionnaire validation [25,27]. To guarantee the clarity of the questions, content and face validity tests were conducted and several changes were made to improve the reliability and scientific value of the data to be collected. The Cronbach’s coefficient reliability test range yielded 0.70–0.95% for each component of the questionnaire (Cronbach’s α = 0.902) and no significant difference was observed in the reliability coefficient between males and females. The questionnaire included twenty-five (25) questions consisting of close-ended and multiple-choice options, pertaining to VD knowledge, behavior, and attitude. The questionnaire had a cover letter (Arabic and English). The final version was uploaded digitally and distributed through WhatsApp groups and Twitter handles. The questionnaire contained five (5) sections: the first part consisted of baseline demographic data (residence, marital status, education, family income, and smoking history); the second part was anthropometric measurements (height, weight, and BMI); the third part, medical history, and health data (if you have VD deficiency will you do a test, taking VD supplement, what was your VD level during past three months, awareness of VD related diseases, enough VD dose, motivation to take VD supplement; fourth part, awareness (awareness of VD sources, people at risk of VD deficiency, the role of VD in body, sunlight exposure, using sun blocker, the optimal time to exposure; and fifth part, physical activity (type, duration, and place).

### 2.3. Statistical Analysis

Data were analyzed using SPSS (version 22, Chicago, IL, USA). Continuous data were presented as mean ± standard deviation (SD) and categorical data were presented as frequencies and percentages (%). All continuous variables were checked for normality using the Kolmogorov-Smirnov test. VD Knowledge, behavior, and attitude on basis of BMI Status were analyzed using Chi-square and fisher exact tests. Multinomial logistic regression was performed for the odd ratio to BMI status with the main source and dietary source of VD knowledge. *p*-value < 0.05 and 0.01 was considered statistically significant.

## 3. Results

### 3.1. Demographic and Anthropometric Information

The demographic and anthropometric information is presented in Table 1. A total of 774 males and females were recruited, and among them, the percentage of OW and OB was 32%, and 18.5%, respectively. Moreover, 49.2% were single and the majority had a university education (54%). The income of 45.6% of participants belong to an average level, while 35.9% had a high income.

### 3.2. Knowledge, Behavior, and Attitude

Table 2 shows the knowledge, behavior, and attitude of participants with respect to VD stratified according to BMI status.

#### 3.2.1. Knowledge about VD

In general, the majority of the participants, including OW (97.2%) and OB (96.5%) were well aware of VD. The majority of OB participants (95.1%) correctly identified sunlight as a VD source, and it was significantly higher than OW participants (90.7%) and controls (88%) (*p* < 0.001). Furthermore, significantly more OW participants answered food and fortified food as VD sources than controls (83.1% versus 81.2% *p* = 0.039; 66.5% versus 58%; *p* = 0.02, respectively). Consequently, 18.9% of OB participants also wrongly identified air as a VD source and this was significantly higher than OW participants (12.9%) and controls (14.4%); *p* = 0.03. For questions related to knowledge about good dietary sources of VD, such as fortified cow’s milk, a significantly higher prevalence of OB participants (43.4%) correctly responded as compared to OW (42.3%) and controls (33.2%). Moreover, the general pattern of correct answers about good dietary sources of VD was high in OW and OB than in controls.

When asked about the role of VD in the human body, the percentage of OB and OW participants with the correct answer (such as developing mineralization and helping to maintain muscle strength) was significantly higher (79.7%, *p* = 0.01, 75.8%, *p* = 0.01, respectively) than controls. A significantly higher percentage of correct responses were also seen from OW participants about diseases linked with VD deficiency such as bone disease (68.5%, *p* < 0.024) than controls (58.2%). Other correct answers for VD-related diseases (inflammatory bowel disease, rickets, and osteoporosis) were not significantly different between groups.

#### 3.2.2. Behaviors and Attitudes about VD

When asked about getting enough sun exposure, the control group responded positively with the highest frequency (31.3%) than OW (25.4%) and OB participants (29.4%). A significant difference (*p* < 0.04) was observed among participants when asked about “which part of your body is exposed to the sun?” Majority of participants responded ‘hand’ (OW (66.5%), control (65.8%) and OB (54.5%)). The controls were more likely to use sun protection during summer and autumn (28%, *p* < 0.003) than OW and OB participants (15.3%, and 14.7%, respectively). The indoor physical activity (PA) was significantly higher among OW participants (29.4%, *p* < 0.01) than among controls (18.7%). Responses about the dose of VD supplements (such as <1000 IU, >1000 IU) and factors such as motivation for initiating VD supplements were insignificant among groups. When asked about “taking supplement is only necessary in case of lack of exposure to sunlight”, affirmative responses from OW participants were significantly higher (63.3%) than OB participants (58%) and controls (52.7%). A significantly higher percentage of OB participants (83.9%, *p* < 0.032) responded positively when asked if they are interested to know more about VD than OW participants (76.6%) and controls (72.8%).

### 3.3. Vitamin D Knowledge and Predictors of BMI Status

Table 3 shows the odds of predicting BMI status based on participants’ VD knowledge using control as a reference. OW participants were 50% less likely to identify salmon and fish oil (Odds ratio, OR 0.5 (95% Confidence interval, CI 0.4–0.7); *p* < 0.01) and 40% more likely to identify chicken (OR 1.4 (1.0–1.9); *p* < 0.05) as dietary sources of VD than controls. On the other hand, OB participants were almost three times more likely to know that sunlight exposure is the main source of VD than controls (OR 2.65 (1.2–6.0); *p* < 0.05). The rest of the odds ratios were not significant (Table 3).

## 4. Discussion

Given that both VD deficiency and obesity are highly prevalent in Saudi Arabia, the present study investigated whether Saudi adults’ knowledge and perceptions about VD are associated with BMI status. The study has clinical merits especially among OW and OB individuals since they are more susceptible to VD deficiency. Proper knowledge of VD dietary sources may favorably affect the individual’s intention to correct VD status and perceived benefits without consuming the added calories that goes along with these foods [28]. In the present study, the majority of the participants have good knowledge about VD independent of BMI status. Although the knowledge about most of the overall sources of VD was correct and significantly higher in OW and OB groups than controls, there was still some misinformation among OB individuals reporting air as one of the overall sources of VD, as well as OW individuals being less likely to know that salmon and fish oil are better sources of VD than chicken which does not only have a lesser amount of VD but also the less healthy option for having substantially smaller amounts of saturated fat. Our above findings align with one study performed among French adults demonstrating various gaps in knowledge by responding to incorrect sources and health effects of VD [29]. In addition, the control group also responded with some wrong answers about good dietary sources of VD such as fruits. The majority of the participants are highly educated (university level) and as explained earlier, rely more on dietary intake sources rather than sun exposure. A study performed among Saudi adults demonstrated confusion and shortage of knowledge as participants responded that eating a variety of fruits and vegetables is a protective way to reduce VD deficiency [23]. The present study confirms the previous findings showing limited knowledge about good dietary sources of VD even among normal-weight individuals. The knowledge about diseases associated with VD deficiency was also higher among OW and OB participants than in controls. A possible reason for more accurate knowledge in these populations reflects their interest to know more about diseases and how it relates to their condition. Similarly, the significant high knowledge among OW and OB groups regarding the role of VD in the body such as developing mineralization, muscle strength, and aid in immune system function suggests that having the right knowledge does not necessarily translate to healthier lifestyles [30,31].

The present study revealed other interesting findings between behavior and attitude related to VD based on BMI status. For instance, a significantly higher proportion of OW (76.6%) and OB (83.9%) participants were curious to know more about VD than controls (72.8%). The present results also showed negative behavior towards sun exposure among OW and OB groups. Although statistically insignificant, a considerable proportion of OW (25.4%) and OB (29.4%) participants agreed that they are not getting enough sun exposure. In addition, the average duration of sun exposure such as 15–30 min is relatively lower among high BMI participants than in controls.

The positive attitude to undergo a test for VD but not exposing themselves to direct sunlight as a measure to reduce the risk for VD deficiency demonstrates a gap in the translation of knowledge. These findings agree with studies showing inconsistencies between attitude and behavior toward sun exposure among Arabic gulf populations and even among Chinese individuals who possess enough knowledge about VD [32,33]. Moreover, it has been demonstrated that attitude is also affected by other variables such as belief [34,35]. As explained earlier, the females who constitute a majority of our participants might be using traditional covering (abaya) which is demonstrated as the most commonly used sun protection in Saudi Arabia [20,21].

The present study showed significantly poor sun exposure behavior among a majority of the OW participants than the control. The OW participants (66.5%) exposed only hands to direct sunlight which represents a very small portion of their bodies. On the other hand, although non-significant, the majority of the control group exposed either their face (54.6%) or both face and hands (64.8%) than OW (60.5%) and OB (57.3%) groups. A study performed on Indian students also showed that the perceived susceptibility of a person towards certain diseases can affect their attitude in following preventive health measures [36].

In the present study, the frequency of physical activity was high among controls, and many preferred outdoor rather than indoor activities. However, the use of sun protection in summer and autumn was also significantly high in this group. In addition, a significantly low proportion of the control group (52.7%) supports that taking a supplement is only necessary in case of lack of exposure to sunlight. These negative behaviors among controls suggest a lack of awareness and that even though they are getting enough sun exposure, they are short of knowledge on how to maximize the benefits when they are exposed to direct sunlight. Our findings corroborate with a Malaysian study demonstrating that although the respondents have adequate knowledge about VD, they showed negative behavior towards sun exposure [37]. Studies from China and Australia also reported negative attitudes toward exposure to sunlight [38,39]. In addition, the same pattern of mixed feelings and inconsistent behavior about sun exposure was reported among adults in Saudi Arabia as well [23].

In our present study, the odds ratio of knowledge about good dietary sources of VD was not associated with BMI status and hence, cannot be considered a strong predictor of BMI. In addition, obese individuals had a higher knowledge with regards to sunlight as the main source of VD than controls. A recent study among the geriatric population in Indonesia demonstrated that OB respondents had better knowledge about sun exposure than non-OB and the odds ratio results showed that good knowledge of sun exposure has the same risk in both groups (OB and non-OB) [39]. Moreover, studies also showed that the negative attitude toward sun exposure among respondents might be due to awareness about harmful health consequences of sun exposure such as skin cancer and aging [40,41].

### 4.1. Limitations

The current study has several limitations. First, the cross-sectional design of the study does not allow to establish the exact causality between the study variables. Second, the information collected is online and self-reported, thus our results might be subject to recall bias or inaccuracy. Third, the questionnaire in the present study does not include the reasons for avoiding sunlight, which could have added more information about the participant’s perception and behavior toward sunlight exposure. In addition, based on remote locations or poor internet availability, it is possible that our online survey might not reach some of the target population.

### 4.2. Strength

Despite limitations, our study has several strengths. There is no such study in Saudi Arabia demonstrating the association of BMI status with VD knowledge, behavior, and attitude among this section of the population. The results of the present study provide valuable information that although the knowledge about VD is high among OB and OW individuals, the quality of knowledge is inconsistent, and this may affect the proper translation of knowledge into their lifestyle. Moreover, the study provides evidence of an individual’s perception playing a role related to BMI status. These results may be used to perform further research in order to explore the underlying motives for negative behavior and shortage of knowledge in specific aspects of VD in the same high-risk population.

## 5. Conclusions

The present study demonstrated a high level of knowledge among Saudi adults about VD irrespective of BMI status, but with limitations in certain areas such as VD sources particularly among OW and OB participants, indicating inconsistencies in the quality of VD knowledge in this population. Public health awareness campaigns should consider addressing some of the wrong perceptions about VD especially among OW and OB individuals since this may adversely affect the proper translation of knowledge in their lifestyle and behavior.

## Figures and Tables

**Table 1 ijerph-19-11107-t001:** Demographic and Anthropometric information of the subjects.

Parameters	All
Sex (M/F)	774 (239/535)
Age (years)	30.8 ± 12.5
BMI (kg/m^2^)	25.6 ± 6.0
BMI Status	
Control	383 (49.5)
Overweight (OW)	248 (32.0)
Obese (OB)	143 (18.5)
Marital Status	
Single	381 (49.2)
Married	352 (45.5)
Widow	18 (2.3)
Divorce	23 (3.0)
Education Status	
University	418 (54.0)
Postgraduate	125 (16.1)
Secondary	180 (23.3)
Primary	43 (5.6)
Read and Write	8 (1.0)
Income Status	
<5000 SAR	143 (18.5)
5001–15,000 SAR	353 (45.6)
>15,000 SAR	278 (35.9)
Smoking Status	
Frequent	43 (5.6)
Occasional	40 (5.2)
Ex-Smoker	32 (4.1)
Never Smoke	659 (85.1)
Family History	
DM	469 (60.6)
HTN	467 (60.3)
Heart Disease	213 (27.5)
Dyslipidemia	174 (22.5)
Cancer	120 (15.5)
Osteoporosis	159 (20.5)
Arthritis	226 (29.2)
Bone fracture history	75 (9.7)
Medical History	
DM	72 (9.3)
HTN	71 (9.2)
Heart Disease	25 (3.2)
Dyslipidemia	45 (5.8)
Cancer	7 (0.9)
Osteoporosis	26 (3.4)
Arthritis	48 (6.2)
Bone fracture history	18 (2.3)

Note: Data presented N (%) and mean ± SD.

**Table 2 ijerph-19-11107-t002:** Vitamin D knowledge, behavior, and attitude on basis of BMI Status information of the subjects.

Parameters	Control	Overweight	Obese	*p*-Value
Sex (M/F)	383 (111/272)	248 (75/173)	143 (53/90)	
**Knowledge about VD**				0.39
Have you ever heard about Vitamin D				
Yes	364 (95.0)	241 (97.2)	138 (96.5)	
No	19 (5.0)	7 (2.8)	5 (3.5)	
**Vitamin D belongs to**				0.83
Fat Soluble Vitamin (Correct)	98 (25.6)	70 (28.2)	34 (23.8)	
Soluble vitamin all solution	30 (7.8)	20 (8.1)	13 (9.1)	
Water Soluble Vitamin	48 (12.5)	33 (13.3)	14 (9.8)	
Don’t know	207 (54.0)	125 (50.4)	82 (57.3)	
**Effects of vitamin D on health is important**				0.4
Yes (Correct)	336 (87.7)	225 (90.7)	131 (91.6)	
No	11 (2.9)	3 (1.2)	4 (2.8)	
Don’t know	36 (9.4)	20 (8.1)	8 (5.6)	
**Overall Sources of Vitamin D**				
Food (Correct)	311 (81.2)	206 (83.1)	117 (81.3)	0.039
Supplements (Correct)	305 (79.6)	200 (80.6)	111 (77.6)	0.29
Exercise	93 (24.3)	51 (20.6)	32 (22.4)	0.43
Air	55 (14.4)	32 (12.9)	27 (18.9)	0.03
Sunlight (Correct)	337 (88.0)	225 (90.7)	136 (95.1)	<0.001
Fortified Food (Correct)	222 (58.0)	165 (66.5)	88 (61.5)	0.02
Water	86 (22.5)	38 (15.3)	32 (22.4)	0.26
**Main Source of Vitamin D**				0.116
Exercise	9 (2.3)	4 (1.6)	1 (0.7)	
Food	31 (8.1)	22 (8.9)	7 (4.9)	
Sunlight (Correct)	315 (82.2)	195 (78.6)	125 (87.4)	
Supplement	22 (5.7)	26 (10.5)	10 (7.0)	
Water	6 (1.6)	1 (0.4)	0 (0.0)	
**Dietary sources sufficient to correct Vitamin D**				0.21
Yes	113 (29.5)	64 (25.8)	33 (23.1)	
No (Correct)	190 (49.6)	142 (57.3)	76 (53.1)	
Don’t know	80 (20.9)	42 (16.9)	34 (23.8)	
**Good dietary sources of Vitamin D**				
Fruits	194 (50.7)	88 (35.5)	65 (45.5)	0.002
Sardine, Salmon, oily fish (Correct)	229 (59.8)	165 (66.5)	88 (61.5)	0.16
Cow’s Milk (Correct)	137 (35.8)	89 (35.9)	53 (37.1)	0.21
Eggs yolks (Correct)	184 (48.0)	140 (56.5)	76 (53.1)	0.09
Chicken (Correct)	76 (19.8)	46 (18.5)	25 (17.5)	0.16
Vegetables	173 (45.2)	98 (39.5)	62 (43.4)	0.45
Infant or toddler formula	153 (39.9)	109 (44.0)	65 (45.5)	0.06
Red Meat (Correct)	81 (21.1)	62 (25.0)	22 (15.4)	0.08
Bread	43 (11.2)	21 (8.5)	9 (6.3)	0.01
Breast Milk	178 (46.5)	133 (53.6)	79 (55.2)	0.09
Fortified Cow’s Milk (Correct)	127 (33.2)	105 (42.3)	62 (43.4)	0.01
Liver	136 (35.5)	79 (31.9)	54 (37.8)	0.08
Others	73 (19.1)	43 (17.3)	16 (11.2)	0.03
**Indoor-working at high risk of Vitamin D**				0.82
Yes (Correct)	328 (85.6)	216 (87.1)	127 (88.8)	
No	15 (3.9)	10 (4.0)	6 (4.2)	
Don’t know	40 (10.4)	22 (8.9)	10 (7.0)	
**Outdoor-working at High risk of Vitamin D**				0.37
Yes	24 (6.3)	17 (6.9)	5 (3.5)	
No (Correct)	290 (75.7)	195 (78.6)	120 (83.9)	
Don’t know	69 (18.0)	69 (18.0)	18 (12.6)	
**Role of vitamin D in the body**				
Absorption of calcium and phosphorous	232 (60.6)	142 (57.3)	85 (59.4)	0.35
Antioxidant	139 (36.3)	94 (37.9)	56 (39.2)	0.33
Developing mineralization	275 (71.8)	193 (77.8)	114 (79.7)	0.01
Aid with Immune System function	251 (65.5)	170 (68.5)	91 (63.6)	0.81
Help to muscles strength	249 (65.0)	188 (75.8)	87 (60.8)	0.01
Needed for blood clotting	86 (22.5)	66 (26.6)	30 (21.0)	0.51
**Season affects the amount of time in the sun to synthesize adequate vitamin D**				0.6
Yes	274 (71.5)	176 (71.0)	100 (69.9)	
No	39 (10.2)	32 (12.9)	13 (9.1)	
Don’t know	70 (18.3)	40 (16.1)	30 (21.0)	
**Diseases associated with low vitamin D**				
Breast cancer	32 (8.4)	22 (8.9)	16 (11.2)	0.59
Skin color	68 (17.8)	47 (19.0)	28 (19.6)	0.87
Type1 DM	71 (18.5)	51 (20.6)	27 (18.9)	0.81
Inflammatory bowel disease (Correct)	49 (12.8)	41 (16.5)	25 (17.5)	0.27
Multiple sclerosis	72 (18.8)	67 (27.0)	27 (18.9)	0.04
Rheumatoid arthritis	125 (32.6)	110 (44.4)	64 (44.8)	0.003
Depression	243 (63.4)	146 (58.9)	92 (64.3)	0.43
Renal Disease	62 (16.2)	46 (18.5)	31 (21.7)	0.33
Bone Disease (Correct)	223 (58.2)	170 (68.5)	94 (65.7)	0.02
Gallstone	42 (11.0)	36 (14.5)	22 (15.4)	0.27
Heart Disease	71 (18.5)	36 (14.5)	26 (18.2)	0.4
Rickets (Correct)	171 (44.6)	119 (48.0)	77 (53.8)	0.17
Osteoporosis (Correct)	242 (63.2)	170 (68.5)	100 (69.9)	0.22
**Motivation for starting vitamin D supplements**				0.16
A friend or family members	26 (6.8)	14 (5.6)	11 (7.7)	
Doctor or health care profession	55 (14.4)	35 (14.1)	32 (22.4)	
Have VD deficiency	209 (54.6)	131 (52.8)	69 (48.3)	
Read about vitamin D on the internet, social media	49 (12.8)	43 (17.3)	17 (11.9)	
School/University	28 (7.3)	21 (8.5)	12 (8.4)	
Seminar/Workshops related to Vitamin D	16 (4.2)	4 (1.6)	2 (1.4)	
**Behavior and attitude about Vitamin D**				0.2
Have you taken a test for Vitamin D				
Yes	207 (54.0)	150 (60.5)	91 (63.6)	
No	70 (18.3)	45 (18.1)	21 (14.7)	
Don’t know	106 (27.7)	53 (21.4)	31 (21.7)	
If yes, are you vitamin D deficient?				0.21
Yes	182 (87.9)	139 (92.7)	84 (92.3)	
No	19 (9.2)	9 (5.0)	4 (4.4)	
Don’t know	6 (2.9)	2 (1.3)	3 (3.3)	
**Taking vitamin D supplements, unless recommended by physician is wrong**				
Yes	220 (57.4)	164 (66.1)	91 (63.6)	
No	85 (22.2)	43 (17.3)	25 (17.5)	
Don’t know	78 (20.4)	41 (16.5)	27 (18.9)	
**Are you getting enough sun exposure?**				0.56
Yes	120 (31.3)	63 (25.4)	42 (29.4)	
No	234 (61.1)	168 (67.7)	91 (63.6)	
Don’t know	29 (7.6)	17 (6.9)	10 (7.0)	
**Length of daily sun exposure**				0.98
<15 min	122 (31.9)	84 (33.9)	47 (32.9)	
15–30 min	193 (50.4)	123 (49.6)	70 (49.0)	
30–60 min	28 (7.3)	21 (8.5)	10 (7.0)	
>60 min	8 (2.1)	4 (1.6)	3 (2.1)	
None	32 (8.4)	16 (6.5)	13 (9.1)	
**Part of your body exposed to sunlight**				
Face	209 (54.6)	127 (51.2)	67 (46.9)	0.27
Hand	252 (65.8)	165 (66.5)	78 (54.5)	0.04
Face and Hand	248 (64.8)	150 (60.5)	82 (57.3)	0.28
Both Arm	247 (64.5)	162 (65.3)	88 (61.5)	0.78
Both Legs	223 (58.2)	139 (56.0)	73 (51.0)	0.21
Completely Cover	120 (31.3)	78 (31.5)	46 (32.2)	0.97
**Walk outdoors daily for sufficient exposure**				0.42
Yes	284 (74.2)	188 (75.8)	114 (79.7)	
No	99 (25.8)	60 (24.2)	29 (20.3)	
**Taking VD supplements only necessary in case of lack of sunlight exposure**				0.037
Yes	202 (52.7)	157 (63.3)	83 (58.0)	
No	109 (28.5)	61 (24.6)	31 (21.7)	
Don’t Know	72 (18.8)	30 (12.1)	29 (20.3)	
**Sun Protection in summer and autumn**				0.003
Always	110 (28.7)	38 (15.3)	21 (14.7)	
Never	105 (27.4)	70 (28.2)	49 (34.3)	
Often	58 (15.1)	46 (18.5)	24 (16.8)	
Rarely	46 (12.0)	43 (17.3)	22 (15.4)	
Sometimes	64 (16.7)	51 (20.6)	27 (18.9)	
**My primary sun protection is**				0.056
Hat	40 (10.4)	32 (12.9)	19 (13.3)	
Sunscreen Cream	195 (51.2)	99 (39.9)	53 (37.1)	
Umbrella	30 (7.8)	29 (11.7)	18 (12.6)	
Nothing	59 (15.4)	52 (21.0)	26 (18.2)	
Other	58 (15.1)	36 (14.5)	27 (18.9)	
**Are you interested to know more about vitamin D?**				0.03
Yes	279 (72.8)	190 (76.6)	120 (83.9)	
No	61 (15.9)	30 (12.1)	8 (5.6)	
Don’t know	43 (11.2)	28 (11.3)	15 (10.5)	
**Physical activities performed**				0.058
Yes	289 (75.5)	180 (72.6)	93 (65.0)	
No	94 (24.5)	68 (27.4)	50 (35.0)	
If yes PA				0.01
Indoor	54 (18.7)	53 (29.4)	19 (20.4)	
Outdoor	132 (45.7)	81 (45.0)	52 (55.9)	
Both	103 (35.6)	46 (25.6)	22 (23.7)	
**Are you walking or exercising outdoors?**				0.14
Yes	62 (16.2)	34 (13.7)	26 (18.2)	
Sometime	127 (33.2)	65 (26.2)	37 (25.9)	
No	194 (50.7)	149 (60.1)	80 (55.9)	
**Ever took supplements or multivitamins that contain Vitamin D**				0.95
(Yes)	183 (47.8)	125 (50.4)	70 (49.0)	
**Vitamin D supplement dose**				0.17
<1000 IU	85 (22.2)	61 (24.6)	28 (19.6)	
>1000 IU	74 (19.3)	61 (24.6)	39 (27.3)	
Don’t know	224 (58.5)	126 (50.8)	76 (53.1)	

Note: Data presented N (%). *p*-Value significant at 0.05 and 0.01 levels using Chi-square and Fisher exact test.

**Table 3 ijerph-19-11107-t003:** Knowledge of VD sources as predictors of BMI status.

Dietary Sources of Vitamin D							
	Foods	Salmon, Oil Fish	Cow’s Milk	Eggs Yolks	Chicken	Red Meat	Fortified Cow’s Milk
BMI Status							
Control	1	1	1	1	1	1	1
OW	0.8 (0.6–1.2)	0.5 (0.4–0.7) **	1.3 (0.9–1.9)	1.3 (0.9–1.9)	1.4 (1.0–1.9) *	1.2 (0.8–1.6)	1.3 (0.9–1.8)
OB	0.7(0.5–1.1)	0.8 (0.6–1.2)	1.1 (0.7–1.6)	1.0 (0.7–1.5)	1.2(0.8–1.8)	1.3 (0.9–1.8)	1.4 (0.9–2.1)
Main Sources of Vitamin D							
	Foods		Supplements			Sunlight exposure	
BMI Status							
Control	1		1			1	
OW	1.14 (0.7–1.7)		1.07 (0.7–1.6)			1.33 (0.8–2.3)	
OB	1.04 (0.6–1.7)		0.90 (0.6–1.4)			2.65 (1.2–6.0) *	

Note: Data presented in odd ratio (95% CI). * & ** represented *p*-value significant at 0.05 and 0.01 level.

## Data Availability

Data is contained within the article.

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
