# Peer review of "Association of Vitamin D Knowledge, Behavior and Attitude with BMI Status among Arab Adults"

_ijerph, 2022, doi:10.3390/ijerph191711107_

Round 1

Reviewer 1 Report

Introduction:

Since the cut-off point of serum vitamin D level is controversial, please define vitamin D deficiency and insufficiency levels referred to in this section

Method

- The title mentioned that the study is among Arab adults but the method stated that consent was also obtained from the parent for ages below 18 (lines 100-101)

- It is not clear what is the study inclusion and exclusion criteria

Results:

- Table 1: I think the data for weight and height is wrong

- Table 2 is pretty long and very hard to read. Perhaps can divide into domains (knowledge, behavior, and practice) in a separate table. The written results for this table are also hard to follow 

- Table 3: why the question on vitamin D sources is the only question to represent vitamin D knowledge?

Study limitation

- Line 267: small sample size was one of the study limitations. However, the calculated sample size is met (lines 96 – 98)

Author Response

Reviewer 1

  1. Introduction: Since the cut-off point of serum vitamin D level is controversial, please define vitamin D deficiency and insufficiency levels referred to in this section.

Response: We thank the reviewer for this comment. Cut-off points were provided and references updated.

  1. The title mentioned that the study is among Arab adults but the method stated that consent was also obtained from the parent for ages below 18 (lines 100-101).

Response: We appreciate the reviewer’s attention to detail. The statement was revised accordingly to clearly reflect that only adults participated in the survey.

  1. Moreover, modify or rephrase the last sentence of materials and methods (2.1. The Study Design and Participants) as it is not correct.

Response: This was similar to the 2nd comment. The statement was revised accordingly to clearly reflect that only adults participated in the survey.

  1. It is not clear what is the study inclusion and exclusion criteria.

Response: Exclusion criteria has been included in the revised subsection 2.1 of the paper. Thank you for pointing this out.

  1. Results: Table 1: I think the data for weight and height is wrong

Response: We thank the reviewer and indeed the 2 values were swapped. The authors decided to remove both parameters in the revised version since the BMI is already included.

  1. Table 2 is pretty long and very hard to read. Perhaps can divide into domains (knowledge, behavior, and practice) in a separate table. The written results for this table are also hard to follow. 

Response: As suggested by the reviewer, table 2 is updated and the questions are replaced based on two domains: (1) Knowledge about VD and (2) attitude and behavior about VD. In table 2, the following questions were moved in the domain of knowledge about VD; “Season affect amount of time in the sun to synthesis adequate VD” and “Motivation for starting Vitamin D supplement”. In the same table, a missing question (diseases associated with VD) which is already discussed in result section is also added in knowledge domain in revised manuscript. As suggested by the reviewer, the written result section has also been divided into different portion, namely (3.2.1.) knowledge about VD and (3.2.2.) attitude and behavior about VD. LINE 143: The following statement is added and rephrased: “Based on their BMI status, Table 2 represents the knowledge, behavior and attitude, among participants”.

  1. Line 188: the section number is 3.3 and not 3.4.

Response: This has been removed.

  1. Table 3: why the question on vitamin D sources is the only question to represent vitamin D knowledge?

Response: We focused on the sources as these parameters were the only cluster that showed significant differences between groups.

  1. Study limitation. - Line 267: small sample size was one of the study limitations. However, the calculated sample size is met (lines 96 – 98)

Response: As suggested by the reviewer, we have deleted the following sentence from the limitation section in our updated manuscript and rephrased the whole section accordingly.

Reviewer 2 Report

Dear Authors,

This study therefore aimed to reveal the relationship among knowledge, attitude, and behavior about vitamin D and their association with BMI status in this high risk ethnic group.

The study was very interesting and it is of scientific interest and in line with the aims of the journal. 

However, there are some issues that should be addressed.

Abstract

The Abstract Section was well written.

Introduction

The introduction defines the purpose of the work and its significance. The current state of the research field was reviewed carefully and key publications cited. 

Line 56. Please discuss and cite: doi: 10.3390/jcm10194578, doi: 10.3390/nu10030375, doi: 10.3390/jcm11164662, doi: 10.1136/jnnp-2018-320199.)

Materials and Methods

-       Line 95: “A total of 774 male and 95 female were recruited.” Please put this information in the Result Section.

-       Line 112: Plese modify “table2” to “Table 2”.

-       Lines 116-117-120-121-122: Please correct the punctuation.

-       Line 163: Please report “P” as”p”

Discussion

This section was well written.

Author Response

Reviewer 2

  1. The Abstract Section was well written.

Response: We thank the reviewer’s appreciation of our work.

  1. The introduction defines the purpose of the work and its significance. The current state of the research field was reviewed carefully and key publications cited. Line 56. Please discuss and cite: doi: 10.3390/jcm10194578, doi: 10.3390/nu10030375, doi: 10.3390/jcm11164662, doi: 10.1136/jnnp-2018-320199.)

Response: We thank the reviewer for suggesting relevant literature to reinforce the multiple associations of VD deficiency in major disorders not previously covered. The recommended studies have been included in the revised introduction. 

  1. Line 95: “A total of 774 male and 95 female were recruited.” Please put this information in the Result Section.

Response: The information is updated in the revised manuscript as suggested by the reviewer.

  1. Line 112: Plese modify “table2” to “Table 2”.

Response: The information is updated in the revised manuscript as suggested by the reviewer.

  1. Lines 116-117-120-121-122: Please correct the punctuation.

Response: The information is updated in the revised manuscript as suggested by the reviewer. The paragraph is rephrased and several punctuations are corrected as shown below:

  1. Line 163: Please report “P” as”p”

Response: The information is updated in the revised manuscript as suggested by the reviewer

  1. This section was well written.

Response: We thank the reviewer for appreciating our work. We have also extensively revised the discussion section taking into consideration the changes in the revised results section. We hope the reviewer will appreciate the changes done.

Round 2

Reviewer 1 Report

Dear authors,

Thanks for the corrections, the manuscript is now clearer than before. My minor comment for line 42 in the abstract: What do you mean by 'mixed feeling' as for the conclusion? I think you may consider to rewrite the sentence. 

Author Response

Comment: Thanks for the corrections, the manuscript is now clearer than before. My minor comment for line 42 in the abstract: What do you mean by 'mixed feeling' as for the conclusion? I think you may consider to rewrite the sentence. 

Response: We thank the reviewer for this comment. We substantially revised the entire abstract as the previous version did not entirely reflect the main findings of the study. The conclusion was also rewritten for added clarity and impact.